# Cyto- and Histopographic Assessment of CPA3-Positive Testicular Mast Cells in Obstructive and Non-Obstructive Azoospermia

**DOI:** 10.3390/cells13100833

**Published:** 2024-05-14

**Authors:** Dmitrii Atiakshin, Nina Kulchenko, Andrey Kostin, Michael Ignatyuk, Andrey Protasov, Ilya Klabukov, Denis Baranovskii, Mikhail Faniev, Elina Korovyakova, Irina Chekmareva, Igor Buchwalow, Markus Tiemann

**Affiliations:** 1RUDN University, 117198 Moscow, Russia; kle-kni@mail.ru (N.K.); andocrey@mail.ru (A.K.); ignatyuk-ma@rudn.ru (M.I.); andrei.protasov@bk.ru (A.P.); faniev.mv@gmail.com (M.F.); elina.korovyakova@mail.ru (E.K.); chia236@mail.ru (I.C.); buchwalow@pathologie-hh.de (I.B.); 2Research Institute of Experimental Biology and Medicine, Burdenko Voronezh State Medical University, 394036 Voronezh, Russia; 3National Medical Research Radiological Centre of the Ministry of Health of the Russian Federation, Koroleva St. 4, 249036 Obninsk, Russiadoc.baranovsky@gmail.com (D.B.); 4Institute for Hematopathology, Fangdieckstr, 75a, 22547 Hamburg, Germany; mtiemann@hp-hamburg.de

**Keywords:** CPA3, mast cells, azoospermia, seminiferous tubules, tissue microenvironment, fibrosis, male infertility

## Abstract

Infertility is an important personal and society disease, of which the male factor represents half of all causes. One of the aspects less studied in male infertility is the immunological testicular microenvironment. Mast cells (MCs), having high potential for regulating spermatogenesis due to fine-tuning the state of the integrative buffer metabolic environment, are one of the most crucial cellular subpopulations of the testicular interstitium. One important component of the MC secretome is proteases that can act as proinflammatory agents and in extracellular matrix (ECM) remodeling. In the testis, MCs are an important cell component of the testicular interstitial tissue (TIT). However, there are still no studies addressing the analysis of a specific MC protease—carboxypeptidase A3 (CPA3)—in cases with altered spermatogenesis. The cytological and histotopographic features of testicular CPA3+ MCs were examined in a study involving 34 men with azoospermia. As revealed, in cases with non-obstructive azoospermia, a higher content of CPA3+ MCs in the TIT and migration to the microvasculature and peritubular tissue of seminiferous tubules were observed when compared with cases with obstructive azoospermia. Additionally, a high frequency of CPA3+ MCs colocalization with fibroblasts, Leydig cells, and elastic fibers was detected in cases with NOA. Thus, CPA3 seems to be of crucial pathogenetic significance in the formation of a profibrogenic background of the tissue microenvironment, which may have direct and indirect effects on spermatogenesis.

## 1. Introduction

Male infertility is a pressing medical and social problem in the modern world, occurring in 5–7% of men [1]. There is an urgent need for male infertility research in general [2]. The proportion of male infertility among married couples reaches 50%, with azoospermia being the most severe form of pathology [3]. Depending on the causal mechanism, non-obstructive azoospermia (NOA) and obstructive azoospermia (OA) are distinguished [4], with a prevalence of 60 and 40%, respectively [5]. In OA, the problem is excretory duct obstruction, with sperm production not affected; by contrast, NOA results from defective spermatogenesis [6]. In cases where the common evaluation tests are negative, azoospermia is named idiopathic, where a final diagnosis involves more detailed investigations to find the cause [7,8]. Identifying the underlying genetic causes, especially in NOA patients, reduces the rate of “idiopathic” cases [9,10]. Seminiferous tubules are the site where spermatogenesis occurs, being formed by supportive and nourishing Sertoli cells and germ cells. Seminiferous tubules are enveloped by a thick basal lamina containing myoid cells (peritubular myoid cells (PTMs)). Seminiferous tubules are integrated in a loose connective tissue named testicular interstitial tissue (TIT). In the extracellular matrix (ECM) of the TIT stand Leydig cells, nerves, and blood vessels, which are responsible for production and delivery of factors that determine the hormonal and immunological control of spermatogenesis. Therefore, fundamental studies of cell subpopulations of the testicular interstitium would expand the potential for investigating new pathogenetic pathways on testicular pathology development. Signaling pathways normally involved in controlling inflammation play fundamental roles in regulating Sertoli cell activity and responses to reproductive hormones, in addition to promoting immune responses within the testis [11]. It is necessary to take into account immunological, paracrine, and endocrine aspects of testicular immune privilege, which is provided by the mechanisms of immune tolerance, local immunosuppression, immunoregulation, and antigen sequestration behind the blood–testis barrier [12]. Of the cells present in the TIT, such as Leydig cells, fibroblasts, macrophages, and lymphocytes, mast cells (MCs) are the least studied. However, MCs are essential in the regulation of local homeostasis and the integrative-buffering metabolic environment. On the one hand, MCs have a diverse repertoire of receptors that provide signaling pathways for selective responses to external and internal stimuli. On the other hand, MCs can highly selectively secrete diverse mediators, such as cytokines and chemokines, to achieve the required effects on the immune and stromal landscapes of specific tissue microenvironments [13,14,15,16,17,18,19,20,21,22]. Specific mast cell proteases, such as tryptase, chymase, and carboxypeptidase A3, are of particular significance in the formation of the proinflammatory and profibrotic background of local tissue niches [23,24,25,26,27,28,29,30]. Recent evidence related MC activity with spermatogenesis defects [31,32,33,34,35,36,37,38,39]. As known, in the organ-specific population of testicular mast cells, there are two subpopulations of mast cells with distinctive phenotypes, interstitial and peritubular, with the latter showing a “fascicular” shape and being integrated in the lamina propria of seminiferous tubules [38,40]. Despite the most crucial biological role of carboxypeptidase A3 in the regulation of the local tissue microenvironment [24,27], the issue of CPA3+ MC participation in the pathogenesis of spermatogenesis has not previously been considered. 

## 2. Materials and Methods

### 2.1. Case Selection

The study included 34 men with azoospermia. The age of the patients ranged from 24 to 52 years with the non-obstructive form and from 26 to 49 years with the obstructive form of male infertility. The samples were obtained from the Center for Reproductive and Cellular Medicine of the State Budgetary Institution Krasnodar City Clinical Hospital. The main criteria for selecting patients was the presence of azoospermia (absence of spermatozoa in the ejaculate), one or more failed attempts to participate in assisted reproductive technology protocols (IVF/ICSI), and the absence of genetic disorders (karyotype changes, microdeletions of the Y chromosome, CFTR), acute inflammatory diseases of the reproductive tract, severe metabolic disorders, systemic diseases, and varicocele. Based on the patients’ clinical examination findings (ejaculate volume, testicular volume, FSH and LH parameters, unaltered vas deferens), all patients were divided into two groups: group 1—patients with NOA (*n* = 22), group 2—patients with OA (*n* = 12). Patients in both observation groups had no history of sexually transmitted infections over the past 10 years. Patients with OA had a history of inguinal hernia repair (*n* = 4, or 33.3%) and mumps in childhood (*n* = 6, or 50%). In addition, four (18.2%) patients with NOA and two (16.7%) patients with OA had a history of blunt closed trauma to the scrotum due to sports activities. No pathologies in the development of the external or internal reproductive organs were detected in all patients during physical examination and diagnostic investigations. All patients involved in the study underwent testicular biopsy using the micro-TESE technique to comply with the assisted reproductive technologies protocol (IVF). If, during native tissue examination, we failed to obtain sperm from one testicle, then a biopsy sample was taken from the contralateral testicle. Using light microscopy, the degree of spermatogenesis impairment in each patient was evaluated on a ten-point scale, where 10 points meant spermatogenesis was completely preserved; 9 meant minor disturbances of spermatogenesis—disorganization of the spermatogenic epithelium, numerous elongated spermatids; 8 meant less than five spermatozoa in the tubule, few late spermatids; 7 meant the absence of spermatozoa and late spermatids, numerous early spermatids; 6 meant the absence of spermatozoa and late spermatids, few early spermatids; 5 meant the absence of spermatozoa and spermatids, numerous spermatocytes; 4 meant the absence of spermatozoa and spermatids, few spermatocytes; 3 meant only spermatogonia; 2 meant the absence of germ cells, only Sertoli cells were present; 1 meant the absence of elements of spermatogenic epithelium (tubular atrophy) [41]. This technique allowed for a semi-quantitative assessment of spermatogenesis. We also assessed disorders of spermatogenesis in the testicle based on qualitative criteria: normal spermatogenesis, hypospermatogenesis, arrest of spermatogenesis, Sertoli cell-only phenotype (SCO), “mixed atrophy”, and complete tubular atrophy. Testicular mast cell subpopulations were divided into interstitial and peritubular, in accordance with the results of previous studies [38,40]. When assessing CPA3+ granules of mast cells, their division into types II and III, depending on the stage of maturity, was taken into account [27].

This study was conducted in accordance with the principles of “The World Medical Association Declaration of Helsinki: ethical principles for medical research involving human subjects” and was approved by the local ethics committee of RUDN University, protocol No. 7, on 22 September 2016. Informed consent was obtained from all patients. All samples were pseudonymized.

### 2.2. Tissue Probe Staining

Tissue probes left over during the routine diagnostic procedure were fixed in buffered 4% formaldehyde and routinely embedded in paraffin. Paraffin tissue sections (5 and 2 µm thick for histochemical and immunohistochemical staining, respectively) were deparaffinized with xylene and rehydrated with graded ethanol according to a standard procedure [42].

### 2.3. Immunohistochemistry and Histochemistry

For the immunohistochemical assay, we subjected deparaffinized sections to antigen retrieval by heating the sections in a steamer with R-UNIVERSAL Epitope Recovery Buffer (Aptum Biologics Ltd., Southampton, SO16 8AD, UK), at 95 °C × 30 min. Blocking the endogenous Fc receptors prior to incubation with primary antibodies was omitted, according to our earlier recommendations [43]. After antigen retrieval and, when required, quenching of endogenous peroxidase, sections were immunoreacted with primary antibodies. The list of primary antibodies used in this study is presented in Table 1.

Immunohistochemical visualization of bound primary antibodies was performed manually, according to the standard protocol [43]. For manually performed immunostaining, primary antibodies were incubated overnight at +4 °C. Bound primary antibodies were visualized using secondary antibodies conjugated with Alexa Fluor-488 or Cy3. Single and multiple immunofluorescence labeling were performed according to standard protocols [42]. The list of secondary antibodies and other reagents used in this study is presented in Table 2.

For the simultaneous detection of tryptase-positive MCs and elastic fibers, a combined protocol was used that included standard immunohistochemical tryptase staining (Table 1) and elastic fiber staining according to Weigert’s method (Table 3).

### 2.4. Controls

Control incubations were: omission of primary antibodies or substitution of primary antibodies by the same IgG species (Dianova, Hamburg, Germany) at the same final con-centration as the primary antibodies. The exclusion of either the primary or the secondary antibody from the immunohistochemical reaction and the substitution of primary antibodies with the corresponding IgG at the same final concentration resulted in a lack of immunostaining. The specific and selective staining of different cells with the use of primary antibodies from the same species on the same preparation was, by itself, sufficient control for immunostaining specificity.

### 2.5. Image Acquisition 

Stained tissue sections were observed on a ZEISS Axio Imager.Z2 equipped with a Zeiss alpha Plan-Apochromat objective 100×/1.46 Oil DIC M27, a Zeiss Objective Plan-Apochromat 150×/1.35 Glyc DIC Corr M27, and a ZEISS Axiocam 712 color digital microscope camera. Captured images were processed with the software programs “Zen 3.0 Light Microscopy Software Package”, “ZEN Module Bundle Intellesis & Analysis for Light Microscopy”, and “ZEN Module Z Stack Hardware” (Carl Zeiss Vision, Jena, Germany) and submitted with the final revision of the manuscript at 300 DPI. Photomicrographs were obtained in some cases with a Nikon D-Eclipse C1 Si confocal microscope based on the Nikon Eclipse 90i.

### 2.6. Quantitative Analysis 

The content of mast cells in the testicular interstitium under diverse types of azoospermia was evaluated after scanning histological sections on a Leica Aperio AT2 scanner. Planimetric analysis to identify the number of MCs per unit area of the testicular tissue, as well as the absolute number of MCs and other cells of the testicular interstitium, was performed using open-source software for digital pathology image analysis, QuPath [44]. 

### 2.7. Statistical Analysis

The results were statistically processed using the Statistica 8.0 software package. Quantitative group findings were compared using the Student’s *t*-test and Mann–Whitney test. The data in the tables were presented in the form M ± SE, where M is the arithmetic mean, SE is the standard error of the mean, and *p* < 0.05 was taken as the level of statistical significance. The close relationship between the presence of mast cells in the testicular interstitium and changes in the spermatogenic epithelium was evaluated using a selective linear correlation coefficient (values from −1 to +1).

### 2.8. Data Availability

The authors declare that all the data supporting the findings of this work are available within the article or from the corresponding author upon reasonable request.

## 3. Results

The average age of patients was 37.2 years in the NOA group and 34.6 years in the OA group, evidencing the lack of significant difference in age between two groups. The duration of infertility in patients with NOA was 6.8 ± 4.7 years and in patients with OA, it was 4.2 ± 3.8. All examined patients had no changes in karyotype, quantitative chromosomal aberrations, microdeletions of chromosome Y, nor mutations of the CFTR gene. The results of the patients’ clinical examination in both groups are presented in Table 4. Thus, in patients with NOA, the volume of ejaculate was significantly increased compared to that in group 2, since in men with OA the patency of the seminal ducts was impaired; FSH levels were increased, evidencing a disruption of the hypothalamic–pituitary axis in the regulation of spermatogenesis; and in 2/3 patients, the testicle consistency was soft, which indirectly indicated severe disturbance of spermatogenesis in the convoluted seminiferous tubules.

Morphological analysis of testicular biopsies demonstrated that spermatogenesis parameters were significantly different in patients with NOA and OA: 2.77 ± 0.71 and 5.83 ± 0.93 points, respectively (*p* = 0.001) (Figure 1a).

Qualitative disorders of spermatogenesis in patients with NOA were more heterogeneous than in patients with OA (Table 5). Almost every second man with NOA had spermatogenesis arrest. In addition, in patients of group 1, SCO was detected in 40.9% of cases. In turn, in patients with OA, disorders were identified in the form of hypospermatogenesis (50%) and spermatogenesis arrest (33.3%). Spermatogenesis was preserved in 16.7% of patients in the second group. 

In the patients of the first group, morphological analysis of native testicular biopsies detected spermatozoa in only 4 (18.1%) men. In the remaining patients with NOA (*n* = 18; 81.9%), the testicular biopsy results were unsatisfactory and no germ cells were detected. In patients of the second group, morphological analysis of native testicular biopsies detected sperm in 10 (83.3%) men. The number of mast cells in 1 mm^2^ of testicular tissue was 45.22 ± 9.08 in patients with NOA and 26.05 ± 10.31 in patients with OA (*p* < 0.05) (Figure 1). Planimetric analysis revealed an uneven distribution of CPA3+ MCs in the testicular interstitium. It was noteworthy that the majority of MCs were localized in the intertubular interstitium of the testicle, regardless of the type of male infertility; much smaller numbers of MCs were localized near the microvasculature or around the convoluted seminiferous tubules (Table 4). Notably, the study of MC localization in the landscape of the tissue microenvironment revealed significant differences under diverse forms of pathology (Table 4). In NOA, a larger proportion of CPA3+ MCs migrated to the perivascular and periductal tissue microenvironment compared with OA (Table 6, Figure 2 and Figure 3).

The present study confirms the presence of two morphological types of MC, elongated and roundish, depending on the localization in the TIT. MCs are elongated when they make part of the basal lamina (BL) of the seminiferous tubules, standing between PTMs, and they are roundish in the remaining TIT (Figure 2, Figure 3 and Figure 4). 

This fact, evidently, increased the area of contact between MCs and myoid cells and affected the blood–testis barrier. MCs located in the intertubular interstitium of the testicle were larger in size and had a rounded shape. The processing of CPA3 in testicular mast cells was similar to the mechanisms of specific protease biogenesis in MCs of other organs. CPA3 was localized predominantly in secretory granules and accumulated to a lesser extent in the cytoplasm (Figure 2a).

CPA3 gradually accumulated in the secretory granules during their maturation and occupied a peripheral intragranular position in mature large granules (Figure 2a). CPA3 exchange could be observed between mature and immature granules, as was clearly visualized with confocal microscopy (Figure 2a). The number of granules in the cytoplasm could vary from single or a few granules of types II or III to completely filling the cytoplasm (Figure 2a–e). For secretion, MCs sometimes used a specific intracellular spatial localization of CPA3-positive granules that actually led to the isolation of part of the cytoplasm and formation of the MC cytoplasm communication with other targets in the tissue microenvironment (Figure 2a). Sometimes, no nucleus was detected in the cytoplasm of MCs filled with CPA3-positive granules (Figure 2d). 

It was evident that such post-cellular structures carried out active secretory activity in relation to tissue targets, including Leydig cells (Figure 2d). In some cases, under NOA, mast cells located in the TIT could form cytoplasmic projections (Figure 2f). In MCs, there were detected different secretory pathways, including pacemaker degranulation and secretion of individual granules into the extracellular matrix (Figure 2h,i).

In NOA cases, several differences in CPA3+ MC cytotopography and histotopography were observed. In addition to the higher content of MCs in the TIT, the data showed a higher degree of CPA3 secretion into the extracellular matrix to the correlating regulated targets, more frequent MC migration to the endothelium of the microvasculature, and secretion of the protease to endothelial cells (Figure 3a).

Moreover, in comparison with OA cases, MCs were more often detected near the wall of seminiferous tubules (Figure 3b–f). There, they actively degranulated into the extracellular matrix (Figure 3b) and migrated into the tissue microenvironment of peritubular cells (Figure 3c), resting in close contact with myoid cells (Figure 3d,e). Sometimes, CPA3+ MCs really acted as intermediaries between Leydig cells and the wall of the seminiferous tubule (Figure 3f). Similar findings were observed for fibroblast interstitial cells (Figure 3g–j), including myofibroblasts (Figure 3h and Figure 5a). In the testicular interstitial tissue, extensive areas filled with autonomous secretory granules were frequently detected, in which CPA3-positive isolated secretory vesicle contents were observed (Figure 3p,q and Figure 5b). 

The findings obtained correlated with the detection of a network of elastic fibers in the TIT and a higher frequency of CPA3+ MC colocalization with the fibrous component of the extracellular matrix in NOA (Figure 3r,s and Figure 5).

In OA, MCs were located in the intertubular region, as in NOA (Figure 4a). However, the number of MCs was significantly lower compared to NOA (Table 4). The secretion of CPA3 also seemed less intense than in NOA cases in the interstitial space (Figure 4a,b). In addition, the numbers of MCs near fibroblasts (Figure 4c,e), capillaries (Figure 4b,d), basal lamina of seminiferous tubules (Figure 4a), and Leydig cells (Figure 4e) were decreased, while maintaining the constitutive level of secretory activity (Figure 4d,e). Besides the common MC migration into the BL of seminiferous tubules (Figure 4g), MCs of atypical shape were also observed in the BL (Figure 4f). MCs were more frequently localized in the superficial layers of the BL of seminiferous tubules (Figure 4i), forming sometimes groups of several cells (Figure 4h). 

Despite the higher relative content of CPA3+ MCs in the intertubular region in OA cases, rarer colocalization was observed with fibroblasts (Figure 4c), elastic fibers (Figure 4j,k), as well as with the total number of elastic fibers in the TIT. Correlation analysis performed to identify the dependence of spermatogenesis disorders on the amount of CPA3+ MCs per 1mm^2^ of the TIT demonstrated that the Spearman correlation coefficient (ρ) in the group of patients with NOA was equal to −0.763, and it was −0.759 in the group of patients with OA. The relationship between the studied parameters in both observation groups was inverse; the closeness (strength) of the connection was high according to the Chaddock scale. Thus, an increased amount of CPA3+ MCs in the TIT was negatively correlated with disturbed spermatogenesis.

## 4. Discussion

Numerous studies performed previously have demonstrated a close relationship between the increased migration of mast cells into the TIT and the degree of spermatogenesis impairment [34,35,36,45]. Carboxypeptidase A3 is one of three specific mast cell proteases, and an acceptable target for determining the size of the intra-organ mast cell population [27]. The results of the study evidence that the cytotopographic features of CPA3 in the TIT do not differ from the features of this specific protease previously demonstrated by our group in the MCs of other organs [27]. The biogenesis and accumulation of CPA3 in testicular MCs support the gradual accumulation of a specific protease in large secretory granules as they mature. We can assume that testicular MCs contain high levels of CPA3, the content of which changes dynamically depending on the conditions of the tissue microenvironment of the testicle, including the formation of azoospermia. The detection of MCs with large secretory granules, in which CPA3 is preferentially localized in the periphery, evidences an active process of protease secretion into the extracellular matrix, during which the protease is removed from intragranular stores and is actively transported through various secretory pathways across the cytoplasm to molecular targets of the extracellular matrix. Based on our results, we can consider that the previously described two groups of mast cells in the TIT, including “interstitial” mast cells located between Leydig cells, around blood vessels, fibroblasts, and near PTM [38,40,46], or included in the BL itself, are the source of CPA3 in the TIT. The release of individual secretory granules into the tissue microenvironment of the intertubular TIT with the formation of separate tissue fields over a large area detected in NOA cases is objective proof of CPA3 participation in the formation of both a pro-inflammatory background and profibrogenic tissue niches. These niches provide favorable inductive conditions for the excessive formation of the fibrous extracellular matrix, including, in particular, elastic fibers. 

Indeed, it has been previously demonstrated that CPA3 participates in inflammatory mechanisms by promoting the activation of pro-inflammatory biologically active substances, peptides, and enzymes, and can, to some extent, determine the modulation of developing immune reactions [24,27]. As demonstrated in our study, the number of CPA3+ MCs in the TIT in patients with NOA exceeded by almost 40% the similar parameter in patients with OA; these findings evidence the existence of mechanisms for increasing the intra-organ MC population during azoospermia, in particular, due to the increased recruitment of MC precursors from the microvasculature [45]. Apparently, MCs themselves can participate in this process, since their number around the capillaries increased in NOA. This was proven by the analysis of the spatial localization of CPA3+ MCs in the landscape of the tissue microenvironment of testicles in patients with NOA, thus demonstrating their increased content in the peritubular or perivascular interstitium by almost 4 times in NOA cases compared with OA cases. Notably, it is necessary to add to the facts described above that there was a higher degree of CPA3+ MC interaction with Leydig cells and fibroblasts in NOA. We observed a large number of interstitial tissue fields infiltrated with CPA3-positive granules, indicating prolongation of the effects on the targets of the tissue microenvironment, even in the absence of MCs (Figure 3p,q,s). Incorporation was observed, with CPA3 secretory activity, of MCs in the BL of seminiferous tubules in association with PTMs. In addition, the features of CPA3+ MC histotopography with close localization to the nuclei of neighboring somatic cells indicated that juxtaposition of the cytoplasmic membrane of MCs with that of neighboring cells was confined to the regions where the nuclei of neighboring cells were close to their cytoplasmic membranes (Figure 2a,c,d,f,g,h, Figure 3k–n and Figure 4e). In particular, these effects were demonstrated for another specific MC protease—tryptase—in the tumor microenvironment [47,48]. 

Because the effects of specific proteases can supplement each other [26,27,49], simultaneous detection of tryptase, chymase, and CPA3 allows for interpretation of the functional activity of each MC. Considering the successful experience of using multiplex IHC in the study of MC proteases in melanoma and systemic mastocytosis, we plan to devote a separate project to this issue for a more accurate understanding of the role of mast cells in the pathogenesis of male infertility [50,51].

Thus, it can be assumed that the data obtained in the study may support one of the explanations; therefore, a large number of patients with damaged spermatogenesis in the group of patients with NOA had no perspective for its complete restoration, which will negatively affect the results of testicular biopsies and assisted reproductive technologies. Chronic inflammation, which is maintained by MCs via CPA3, contributes to the suppression of spermatogenesis in NOA patients. Men with NOA are thus forced to turn to assisted reproductive technologies (IVF/ICSI, up to the use of donor sperm) more often than men with OA in order to be able to fertilize their spouse’s oocyte [52]. Since there has been no previous study of the correlation between spermatogenesis and CPA3+ MCs, the data of the present study are the first evidence of an inverse relationship between the number of MCs in the TIT and the degree of spermatogenesis impairment.

Peritubular myoid cells and Leydig cells are known to be two significant sources of testicular CCL2 in cases with spermatogenic dysfunction [45]. The chemokine (C-C motif) ligand 2 (CCL2) is also referred to as monocyte chemoattractant protein 1 (MCP1) and small inducible cytokine A2. CCL2 is a small cytokine that belongs to the CC chemokine family. CCL2 tightly regulates cellular mechanics and thereby recruits monocytes, memory T cells, and dendritic cells to the site of inflammation produced by either tissue injury or infection [53]. CCL2 production can evidently explain the higher frequency of CPA3+ MC colocalization with Leydig cells and PTMs when spermatogenesis is impaired. As is known, MC histamine can influence steroidogenesis in Leydig cells by acting through H1R and H2R receptors [31,54,55]. In laboratory animals, proliferation and differentiation of MCs and Leydig cells were demonstrated to occur simultaneously in rat testes, suggesting a dynamic relationship between the two cell types through their secretory products [56]. Undeniably, in the case of NOA, it is necessary to activate the hormone-producing activity of Leydig cells in order to intensify spermatogenesis. Since testicular MCs are the main source of locally produced histamine [54], the evidence of increased MC colocalization with Leydig cells that we have found indicates the activating influence of MCs both through histamine and, probably, CPA3. Concurrently, other secreted MC mediators can contribute to changes in the state of the integrative buffer metabolic environment and provoke fibrosis of the TIT. We believe that these conditions can result in the formation of a vicious circle in which the rational actions of MCs to activate the hormone-producing activity of Leydig cells can be accompanied by the provocation of a local inflammatory reaction and formation of profibrogenic tissue niches. In further research, we will take into account the study of possible interactions of MCs with other resident and non-resident immune cells, providing a complex immune micro-environment, including tissue-resident interstitial and peritubular macrophages, dendritic cells, and lymphocytes [57]. Moreover, an experiment involving laboratory mice with simulated autoimmune orchitis demonstrated a close connection between the developing fibrosis and the activity of macrophages [58]. In addition, the importance of further research dealing with the relationship between the pathogenesis of male infertility and the condition of mast cells was supported in a clinical study that demonstrated the positive effect of 12-week daily use of an antihistamine-like drug with a mast cell-stabilizing effect, reporting a significant increase in morphologically normal sperm cells and significant improvement in sperm motility [59]. Thus, further attention to the biology of MCs in NOA can become a source of new effective solutions in personalized medicine.

## 5. Conclusions

Currently, the fundamental mechanisms that have negative effects on the qualitative and quantitative parameters of spermatogenesis in non-obstructive azoospermia have not been fully studied. The active CPA3 participation in the regulation of the tissue microenvironment of the testicular interstitium detected in the study provides a new perspective in understanding the mechanisms of developing spermatogenesis disorders. Further studies will allow more accurate determination of the involvement of CPA3 in key parts of the pathogenesis of male infertility and may become a promising aim for targeted therapy, including assessment of the interaction of a specific protease with other immune cells of the testicular interstitium. 

## Figures and Tables

**Figure 1 cells-13-00833-f001:**
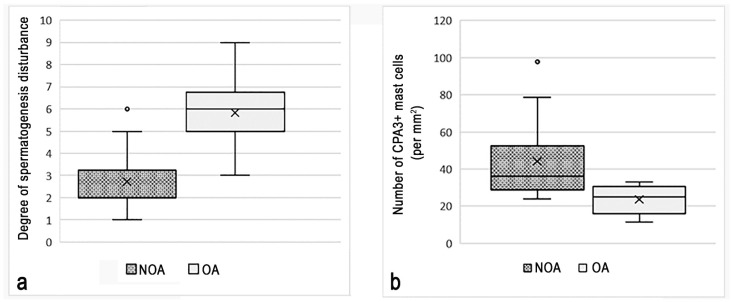
Relationship between the content of CPA3-positive mast cells in the testicular interstitium and the level of spermatogenesis impairment. (**a**) Comparison of the degree of spermatogenesis impairment in patients with non-obstructive (NOA) and obstructive (OA) azoospermia. (**b**) Different content of mast cells in the testicular interstitium in patients with non-obstructive (group 1) and obstructive (group 2) azoospermia.

**Figure 2 cells-13-00833-f002:**
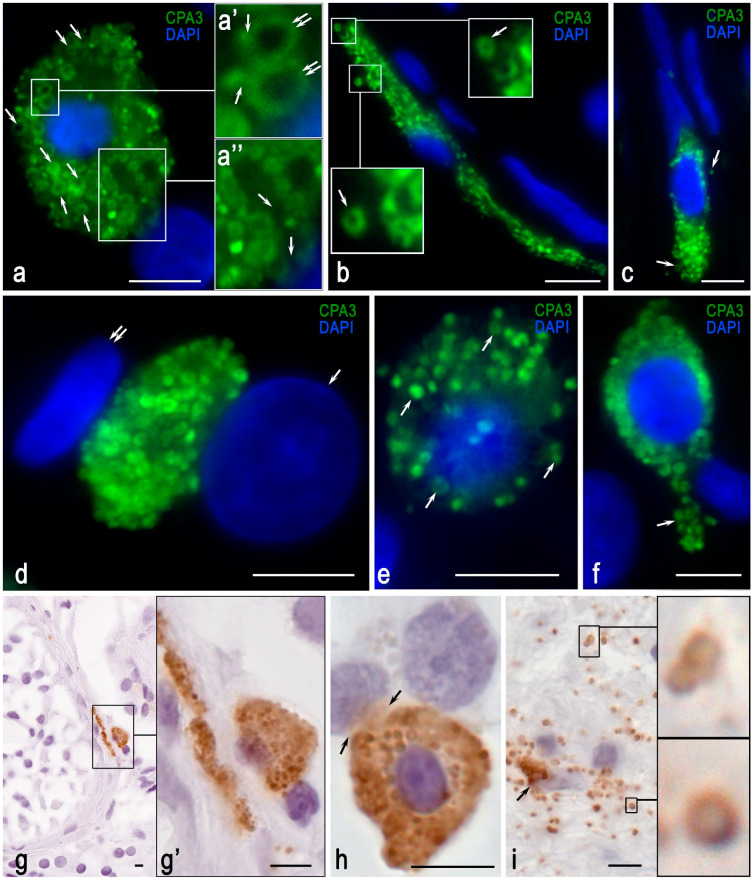
Cytotopographical features of CPA3-positive mast cells in the landscape of the tissue microenvironment of the testicular interstitium. (**a**–**f**) Immunofluorescence images (green: CPA3; blue: nuclei stained with DAPI). (**g**–**i**) Immunohistochemical detection of CPA3. (**a**) Predominant localization of CPA3 in mature mast cell granules (arrowed). (**a’**) Enlarged fragment of (**a**). Exchange of CPA3 between mature and immature granules (arrowed), CPA3 localization along the periphery of mature granules (double arrowed). (**a”**) Enlarged fragment of (**a**). Cooperation of numerous granules forming the secretory cell orifice toward the target cell of the tissue microenvironment (arrowed). (**b**,**c**) Elongated mast cells within the basal lamina of the seminiferous tubule. The secreted material in the extracellular matrix (arrows) can be observed. (**d**) An anucleated fragment of the mast cell cytoplasm juxtaposed to a Leydig cell (arrow) and a myoid peritubular cell (double arrow). (**e**) Low content of CPA3+ granules in the cytoplasm, mostly immature (arrowed), low secretory activity. (**f**) Formation of a mast cell cytoplasmic projection filled with CPA3-positive granules (arrow). (**g**) Observation of the two types of mast cells in the TIT: elongated (left) and round (right). (**g’**) Enlarged fragment of (**g**). (**h**) The mast cell is filled with CPA3-positive secretory granules; targeted protease secretion to the cells of the TIT via the pacemaker degranulation mechanism is evident (arrows). (**i**) CPA3 secretion into the TIT as part of autonomous granules retaining their structure at a great distance from the mast cell (arrowed). Scale bar: 5 µm.

**Figure 3 cells-13-00833-f003:**
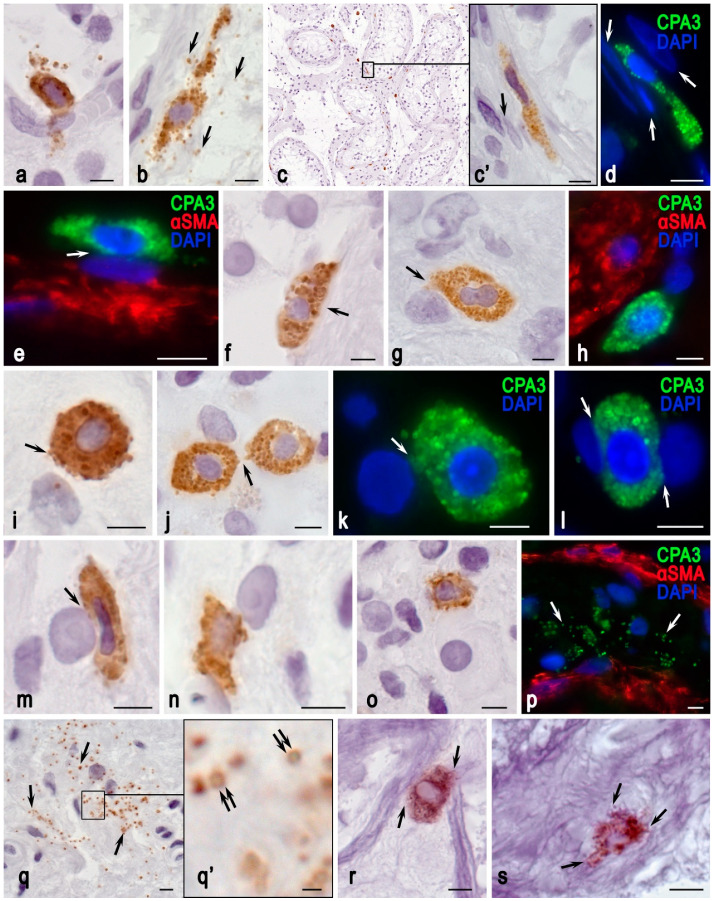
Histotopographic features of CPA3-positive MCs in non-obstructive azoospermia. Techniques: monoplex (**a**–**d**,**f**,**g**,**i**–**o**,**q**) and multiplex (the rest) immunohistochemical detection of CPA3 with simultaneous staining of αSMA (**e**,**h**,**p**), and elastic fibers with Weigert staining (**r**,**s**). (**a**) A pericapillary MC in a state of active CPA3 secretion. (**b**–**f**) Diverse variants of MC localization in the wall of convoluted seminiferous tubules: active CPA3 secretion (arrowed) (**b**), close localization to peritubular cells (arrowed) (**c**,**c’**), association of several myoid cells (arrowed) (**d**), close localization to the nucleus myoid cell (arrowed) (**e**), and wall of the seminiferous tubule (arrowed) (**f**). (**g**–**i**) MC interaction with fibroblasts of the testicular interstitium, with secretory activity (arrowed) (**g**,**i**). (**j**) MC interaction with each other (arrowed). (**k**–**o**) MC interaction with Leydig cells. (**k**) CPA3 secretion to the nucleus (arrowed). (**l**,**m**) Close colocalization with the nucleus without visible signs of secretory activity (arrowed). (**n**) Protease secretion to the Leydig cell nucleus. (**o**) Active CPA3 secretion to the cytoplasmic membrane of the Leydig cell. (**p**,**q**) A large number of CPA3-positive secretory granules in the testicular interstitium (arrowed), in some of them peripheral localization of CPA3 is preserved (double arrowed). (**r**,**s**) Active MC participation in the remodeling of elastic fibers (arrowed). Scale bar: 1 µm (**q’**), 5 µm (the rest).

**Figure 4 cells-13-00833-f004:**
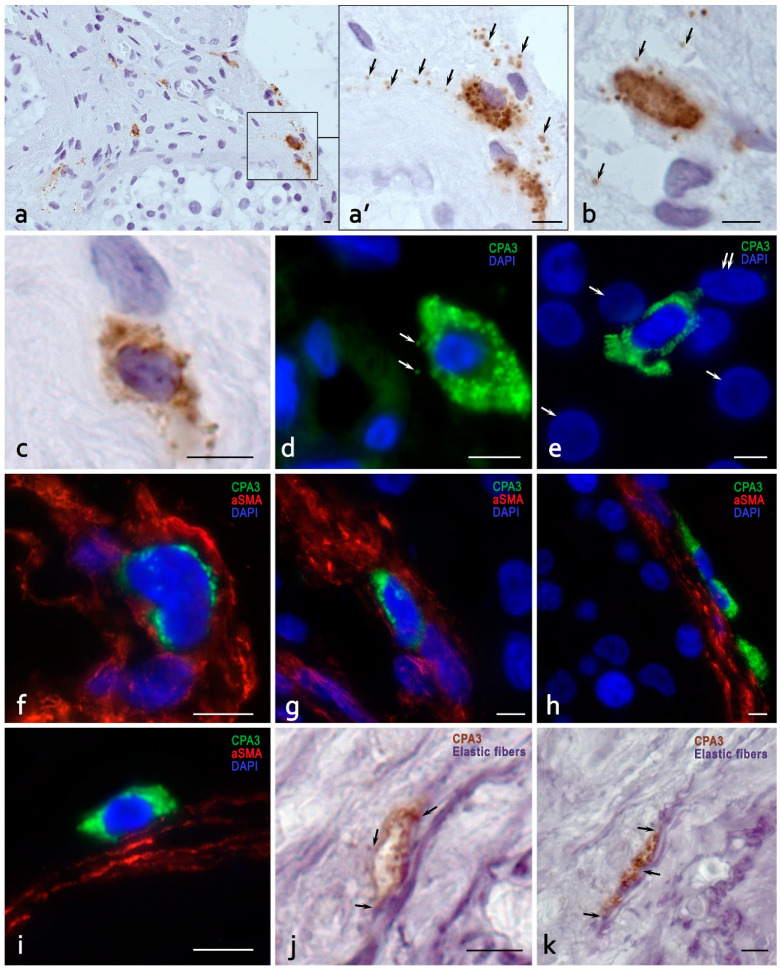
Morphological features of CPA3-positive mast cells in the obstructive form of azoospermia. Techniques: (**a**–**e**) Monoplex IHC of CPA3 mast cells. (**f**–**k**) Multiplex imaging of CPA3 and aSMA (**f**–**i**), CPA3 and elastic fibers (**j**,**k**). (**a**) Predominant localization of mast cells in the interstitium of the testis between the convoluted seminiferous tubules. (**a’**) Enlarged fragment of (**a**). The arrow indicates secretory granules diffusely located in the intercellular matrix of the connective tissue of the testicle. (**b**) MCs during remodeling of the extracellular matrix of the testicular interstitium. The localization of autonomous secretory granules around an MC (arrowed) (**c**) A mast cell interacts with a fibroblast-like cell. (**d**) Perivascular localization of MCs, with CPA3-positive granule secretion to the basement membrane of the endothelium (arrowed). (**e**) A mast cell in the testicular interstitium surrounded by Leydig cells (nuclei are arrowed) and fibroblasts (nuclei are double arrowed). Directed secretion of CPA3 to targets in the tissue microenvironment is visualized. (**f**,**g**) Diverse options for morphological integration of MCs into the layer of myoid peritubular cells of the seminiferous tubules, including internal (**f**,**g**) and external (**h**,**i**) localization. (**j**,**k**) Morphological signs of elastic fiber remodeling with the participation of mast cells (arrowed). Scale: 5 µm.

**Figure 5 cells-13-00833-f005:**
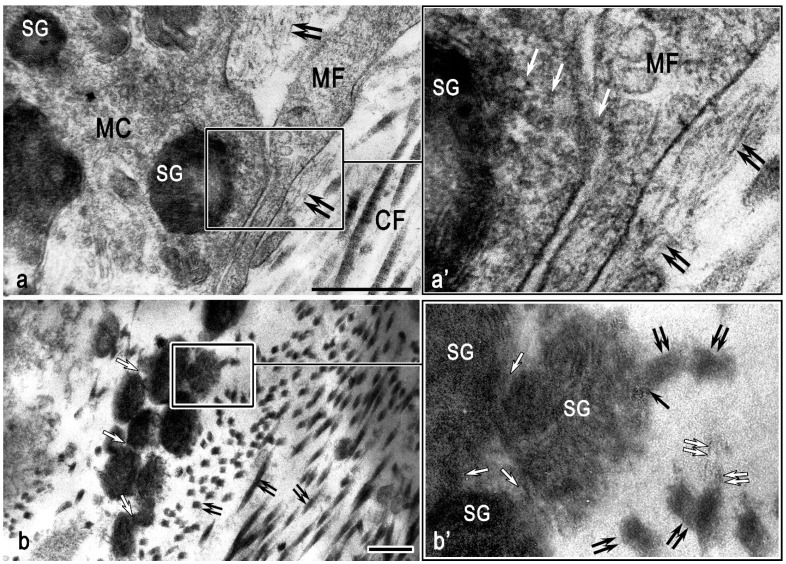
Electron microscope features of mast cell interaction with the structural components of the testicular interstitium in the non-obstructive form of azoospermia. (**a**) Cooperation of a mast cell (MC) and a myofibroblast (MF) in extracellular matrix remodeling. The targeted transport of mast cell secretome components from the secretory granule (SG) (white arrowed), collagen microfibrils (double black arrowed), and mature collagen fibers (CFs) is visualized. (**a’**,**b’**) Enlarged fragment of (**a**,**b**). (**b**) Autonomous secretory granules of mast cells (SGs) in the extracellular matrix. Intergranular exchange (white arrowed), localization of secretory granules in the microenvironment of collagen fibers (double black arrowed), including contact with them (black arrowed). Microfibril polymerization and incorporation into collagen fibers is detected (double white arrowed). Scale: 1 µm.

**Table 1 cells-13-00833-t001:** Primary antibodies used in this study.

Antibodies	Host	Catalogue Nr.	Dilution	Source
Carboxypeptidase A3 (CPA3)	Rabbit polyclonal Ab	#ab251696	1:2000	AbCam, Cambridge, United Kingdom
α SMA	Mouse monoclonal [1A4]	ab7817	1:2000	AbCam, United Kingdom

**Table 2 cells-13-00833-t002:** Secondary antibodies and other reagents.

	Source	Dilution	Label
Goat anti-mouse IgG Ab (#ab97035)	AbCam, United Kingdom	1/500	Cy3
Goat anti-rabbit IgG Ab (#ab150077):	AbCam, United Kingdom	1/500	Alexa Fluor 488
AmpliStain™ anti-Rabbit 1-Step HRP (#AS-R1-HRP)	SDT GmbH, Baesweiler, Germany	ready-to-use	HRP
4′,6-diamidino-2-phenylindole (DAPI, #D9542-5MG)	Sigma, Hamburg, Germany	5 µg/mL	w/o
VECTASHIELD^®^ Mounting Medium (#H-1000)	Vector Laboratories, Burlingame, CA, USA	ready-to-use	w/o
DAB Peroxidase Substrate Kit (#SK-4100)	Vector Laboratories, Burlingame, CA, USA	ready-to-use	DAB

**Table 3 cells-13-00833-t003:** Reagents used for histochemical staining.

Dyes	Catalogue Number	Provider	Dilution	Manufacturer
Mayer’s hematoxylin	HK-G0-DL01	Biovitrum	Ready-to-use	ErgoProduction LLC, Saint Petersburg, Russia
Eosin Y 1% aqueous	HK-EV-A250	Biovitrum	Ready-to-use	ErgoProduction LLC, Russia
Weigert for elastic fibers	21-030	Biovitrum	ready-to-use	ErgoProduction LLC, Russia

**Table 4 cells-13-00833-t004:** Laboratory and clinical parameters of patients of both groups.

Type of Investigation	Parameter of Investigation	Group (NOA) (*n* = 22)	Group 2 (OA) (*n* = 12)
Spermogram (WHO, 2010)	Ejaculate volume	4.2 ± 1.8 *	1.3 ± 0.5
	pH	7.2 ± 0.1	7.4 ± 0.2
	Sperm concentration (mln/mL)	0	0
	Leukocytes (mln/mL)	0.5 ± 0.1	0.4 ± 0.1
	Ig A (%) **	8.1 ± 1.2	8.4 ± 1.1
	IgG (%) **	17.2 ± 2.8	15.3 ± 2.4
Blood hormones	FSH (mlU/mL)	18.4 ± 3.1 *	7.8 ± 4.4
	LH (mlU/mL)	7.3 ± 2.9	5.8 ± 3.2
	Prolactin (ng/mL)	8.5 ± 4.2	9.1 ± 4.6
	Total testosterone (ng/mL)	12.8 ± 3.3	14.2 ± 3.8
Testicular volume (cm^3^)	right	13.8 ± 2.1	15.7 ± 3.4
	left	12.1 ± 3.5	14.2 ± 3.2
Testicle consistency	normal	8 (36.3%) *	12 (100%)
	soft	14 (63.7%)	-

Note: * *p* < 0.05 if compared with parameters in obstructive azoospermia; ** The percentage of IgG and IgA content in the ejaculate were determined using an indirect MAR test for seminal anti-sperm antibodies.

**Table 5 cells-13-00833-t005:** Histopathological disorders of spermatogenesis in patients with NOA and OA.

Qualitative Disorders of Spermatogenesis	Group 1 (NOA) (*n*; %)	Group 2 (OA) (*n*; %)
Normal spermatogenesis	-	2 (16.7%)
Hypospermatogenesis	-	6 (50%)
Arrest of spermatogenesis	10 (45.4%) *	4 (33.3%)
Sertoli cell-only phenotype	9 (40.9%)	-
“Mixed atrophy”	2 (9%)	-
Complete tubular atrophy	1 (4.7%)	-

Note: * *p* < 0.05 if compared with parameters in obstructive azoospermia.

**Table 6 cells-13-00833-t006:** Quantification of CPA3-active mast cells in the testicular interstitium in men with non-obstructive and obstructive azoospermia.

MC Histotopography	Parameter of MC Content	Groups of Patients	
		Non-Obstructive Azoospermia	Obstructive Azoospermia
Intertubular interstitium of the testis	Absolute number per 1 mm^2^	27.64 ± 9.31 *	19.09 ± 2.73
	Relative number among other interstitial cells	0.51 ± 0.12 *	0.32 ± 0.06
Peritubular microenvironment	Absolute number per 1 mm^2^	7.36 ± 1.27 *	1.94 ± 0.05
	Relative number among other interstitial cells	0.13 ± 0.08 *	0.03 ± 0.02
Perivascular microenvironment	Absolute number per 1 mm^2^	10.22 ± 2.69 *	2.62 ± 0.08
	Relative number among other interstitial cells	0.18 ± 0.10 *	0.05 ± 0.03

Note: * *p* < 0.05 if compared with parameters in obstructive azoospermia.

## Data Availability

All data and materials are available upon reasonable request. Address to D.A. (email: atyakshin-da@rudn.ru).

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
