# Peer review of "Cyto- and Histopographic Assessment of CPA3-Positive Testicular Mast Cells in Obstructive and Non-Obstructive Azoospermia"

_cells, 2024, doi:10.3390/cells13100833_

Round 1
Reviewer 1 Report
Comments and Suggestions for Authors
Mast cells (MC) were identified as regular components of the male genital tract including the testis. Increased numbers of MC have been described in testicular biopsies from infertile patients compared to the non-affected, disease-free organ already in the early 1980ies, followed by a series of papers including analysis of MC subsets and their putative function. In their manuscript, Atiaskin et al. re-visit this issue with a special focus on testicular tissue specimens obtained from infertile men suffering azoospermia, choosing CPA3 as MC marker, which is detected by means of immunohistochemistry/-fluorescence. The results represent an interesting piece of information in the field of testicular immunopathology.
There are, however, shortcomings and major points of criticism concerning the manuscript:
Selection of patients, their clinical characterization, and resp. exclusion criteria remain somewhat unclear. Did all azoospermic patients undergo surgery for sperm retrieval, or were biopsies for mere diagnostic purposes included? Did patients undergo bilateral, multi-focal biopsy? If yes, were tissue specimens from any site randomly used for further analysis? What were the criteria for the assignment of patients to the subgroups with non-obstructive azoospermia (NOA) or obstructive azoospermia (OA)? By definition, detection of elongated spermatids / successful sperm retrieval should be 100% in OA; the detection rate in NOA appears fairly low (in published datasets approx. 50%; see Corona et al. 2019) (see results, line 167-171).
Apart from age differences between these subgroups, specific data concerning the medical history (e.g. history of maldescensus testis, previous STI etc.), andrological examination (esp. testicular volume; absence of vas deferens etc.), hormone tests, and results of semen analysis (volume, pH, biochemical parameters such as fructose, alpha-glucosidase) should be compiled and presented with descriptive statistics in a suppl. table. Was semen analysis performed according to WHO recommendations? With regard to genetic testing, mere karyotyping does not represent the “state of the art” to unravel underlying genetic causes, especially for NOA (see Wyrwoll et al. 2022).
In contrast to the Johnsen score (applied in other publications on testicular MC), the semi-quantitative scoring of spermatogenesis (and resp. disorders) proposed by Aafjes et al. is not widely used. What was the rationale for this choice? For statistical analysis, a non-parametric approach is recommended (median [range or interquartile range]). For the assessment of MC in testicular pathologies, diagnostic categorization is mandatory, esp. for NOA patients (Sertoli-cell-only phenotype (SCO), hypospermatogenesis, spermatogenic arrest, “mixed atrophy”, complete tubular atrophy; heterogeneity of findings within or between testes; see McLachlan et al. 2007). This information is completely missing.
The authors focussed on CPA3 as MC marker and high-resolution cyto-morphological details in their IHC and IF analyses. Although there are publications available showing tryptase+ and tryptase+/chymase+ MC in testicular pathologies (see above), a comparative analysis using these markers would corroborate the data obtained here with CPA3. Moreover, representative photomicrographs / histological overviews (e.g. conventional staining such as Giemsa’s plus IHC for CPA3 or tryptase) of specimens with different pathologies would facilitate interpretation of the results compiled in table 4). Additional photomicrographs / histological overviews indicating the areas of interest used for further analysis are also needed in figs. 2-4 (so far only available in fig. 4 a/a’). Did all tissue sections used for IF / IHC in figs. 2-4, resp., were obtained from a single representative specimen or different biopsies?
The differentiation and phenotypic characterization of interstitial and peritubular MC – the latter showing a “fascicular” shape and being integrated in the lamina propria of seminiferous tubules – is not new (see Jezek et al. 1999; Meineke et al. 2000). This available knowledge should be highlighted in the introduction; the approach of phenotypic characterization of MC used in the work presented here needs to be defined in the methods section.
Linking the overall histopathological diagnosis (see above) to the number and phenotypic characterization of MC is mandatory: Highest MC numbers in testis biopsies from infertile men are seen with SCO phenotypes; thus, differential results concerning "OA" and "NOA" presented here are largely biased by the percentage of SCO tubules in respective sections. Moreover, Leydig cell hyperplasia often accompanies severe tubular damage, I.e. SCO. Finally, the degree of fibrosis (e.g. thickening of the lamina propria) is associated with the “shift” from interstial to peritubular localization of MC. The authors should re-analyze their data, taking these aspects into account.
Conclusions drawn from merely descriptive morphological data with regard to a key role of MC in the pathogenesis of testicular damage seen among azoospermic men remain speculative. Apart from Leydig and peritibular cells, the possible interaction of MC with other resident and non-resident immune cells, providing a complex immune micro-environment should be addressed. The authors may refer to recent functional data (although these were largely obtained from animal experiments and refer to macrophages; e.g. Bhushan et al. 2020; Peng et al. 2022). Finally, there are some clinical studies available, though not adequately controlled, targeting MC in male infertility (e.g. Oliva et al. 2006).
Considering the above mentioned aspects, the discussion and conclusions of the paper need to be revised; the text could be shortened. The abstract has to be amended accordingly.
Further details:
Use the established abbreviation “NOA” for non-obstructive azoospermia throughout the text and fig. legends
Line 15: First sentence of the abstract appears inappropriate to introduce the putative role of MC as resident immune cells in testicular dysfunction / disorders of spermatogenesis / male infertility
Lines 15 & 34: male infertility is not “acute” or “pressing”; indeed, however, there is an urgent need for male infertility research in general (see Rimmer et al. Hum Reprod Open, 2022)
Line 38: “prevalence”, not “incidence”
Line 42: identifying underlying genetic causes, esp. in NOA patients, reduces the rate of “idiopathic” cases (see Houston et al. 2021; Wyrwoll et al. 2022)
Lines 45-61: better introduce current knowledge of resident and non-resident testicular immune cells under physiological and pathological conditions (see extensive reviews provided by M. Hedger and others)
Line 75: “late” = “elongated” spermatids
Line 86: pseudonymized
Lines 102-107: sentences unclear, information given somewhat redundant
Line 136-137: re. differential diagnosis of NOA, see comments above
Line 157: measuring IgG and IgA levels in semen is not part of WHO recommended procedures; please specify
Line 176: inappropriate, see comments above
Line 182 (table 4): “Quantification…”
Line 218: “granules of types II or III” needs to be explained (e.g. not mentioned in methods)
Line 224: what does “postcellular structures” mean?
Line 224: “secretory pathways” implicates knowledge about underlying molecular mechanisms; here, it’s merely morphological features of spatial localization of MC and their granules
Line 238: omit “paracrine”; replace “sustentocytes” by “peritubular cells” (see work of M. Mayerhofer and his group) (see also line 256)
Lines 324: phenotypic characterization of MC was published much earlier than in ref [40]; see above
Fig. 1: instead of “group 1” etc. indicate “OA” and “NOA”; replace Y axis legend in A (in Russian); y axis legend in B “No. of…”
Fig. 2: a’), a’’) not indicated in photomicrographs; in i), arrow pointing to MC would be helpful
Fig. 3: add indices for colours used for different antigens in IF in photomicrographs (as in fig. 2)
Tables 2 & 3: could be moved to suppl. materials
Language: editing by native speaker recommended, in order to facilitate reading
Reviewer 2 Report
Comments and Suggestions for Authors
Paper authored by Dr. Atiakshin and colleagues compared human testicular histomorphometry of CPA3-positive mast cells between 12 men with obstructive azoospermia (OA) and 22 men with non-obstructive azoospermia (NOA). They showed NOA men had a high content of CPA3 positive MCs in the testicular interstitial space, and more frequent migration to the microvasculature and the peritubular region of seminiferous tubules as compared to OA testes. They conclude that the presence of high number of MCs in NOA testes may contribute to the pathological changes in NOA testes including spermatogenesis suppression or depletion.
Questions and Comments:
1. Authors provided high quality histological, immunohistological, and EM imagines in supporting their observation and quantification of increased number of MCs in human NOA testes; and provided morphological evidence indicating MCs released CPA3+ granules into testicular interstitium that may cause inflammatory and fibrosis. Do authors have other molecular markers to show testicular inflammation and fibrosis in NOA testes besides increased number CPA3+ MCs and morphological signs of elastic fiber remodeling? This paper appears to be a good descriptive study.
2. It would be more consist with the literature if authors use NOA instead of NA for Non-obstructive azoospermia throughout the text.
3. In 2.7, suggest using M±SE for Mean±Standard error, instead of using M±m.
4. In Figure 1A, please use English for Y-axis description.
Reviewer 3 Report
Comments and Suggestions for Authors
Please see attached comments

Reviewer 4 Report
Comments and Suggestions for Authors
The main question is the potential role of the MCs in male infertility pathogenesis. I consider original the structure of the paper: very simple but effective in order to clarify the potential role of the MCs in non azospermic men. I agree with authors in considering very important to study the intratesticular microenvironment, a very important area for the communication between intratubular and extratubular compartment. In my opinion the paper materialized what has been only theoretical until now on MCs role in male infertility and it has contributed to draw a new practical path on the infertility knowledge: the results strongly suggest a MCs role in this pathology. The paper tracked Mcs in interstitium: periductal and microvasculature area. This findings could open new perspectives on infertility knowledge: migration of MCs toward the interstitium, where they by degranulation produce an inflammatory response with consequent fibrosis. As far as I am concerned, I think that the usefulness of the scientific papers should be of giving clinical practical suggestions. In consideration of this, I suggest to correlate the mean MCs value with the histological categories of male infertility: hypospermatogenesis (mild, medium and severe if possible), complete ed incomplete spermatogenic arrest, Sertoli Cell only Syndrome. In addition I suggest to evaluate if the Leydig Cells are hiperplastic. Certainly we must take in consideration that the case number is low and a wider study is needed, notwithstanding the conclusions addressed us toward a rational way in order to study male infertility. The immunohistochemistry analysis revealed not only an increased MCs number but also their localizzation.
I found the tables and the figures very clear and very well described. And the references are appropriate.
Author Response
The authors are sincerely grateful to the reviewer for this high assessment of our study and supporting the idea of analyzing the tissue microenvironment of the testicular interstitium to form new hypotheses for the formation of male infertility. We will definitely take into account your practical suggestions to improve the translational value of our basic research. We will definitely try to correlate data on the severity of spermatogenesis disorders - hypospermatogenesis (mild, moderate and severe, if possible), complete incomplete arrest of spermatogenicity, Sertoli syndrome only) with the number of mast cells and try to identify a correlation between parameters of reproductive function and the number of mast cells. Qualitative changes in spermatogenesis are presented in Table 5.
Round 2
Reviewer 1 Report
Comments and Suggestions for Authors
In their revision, Atiaskin and co-authors addressed the comments and questions raised during the first review process. Although the rebuttal gives extensive answers in ‘general’ terms, and both, introduction and discussion improved with re-writing, major issues of criticism remain unsolved.
With regard to patients’ clinical and histopathological characterization, essential information is now included. However, specific information concerning the medical history (e.g. history of maldescensus testis, previous STI etc.) is still missing. Interpretation of data shown in Table 4 is partially incorrect: semen volume in OA patients is significantly reduced (surprisingly, pH is normal and no CFTR mutations were found in this subgroup…), compared to normal values in NOA patients. It still remains completely unclear, what IgG and IgA percentages reflect here (results of indirect MAR test for seminal anti-sperm antibodies?). The number of NOA and OA patients should be included in Table 4.
The authors are obviously not familiar with the current definitions for qualitative categorization of disorders of spermatogenesis; data now compiled in Table 5 are largely unplausible. For example, the category “normal spermatogenesis” is missing (to be expected in OA). “Spermatogenic arrest”, esp. when homogeneous, means that there are NO elongated spermatids; so, this diagnosis is incompatible with the assignment of patients to the OA subgroup, the expected prevalence in NOA patients is much lower (refer to reviews, e.g. McLachlan et al. 2007, and original data published e.g. by Tüttelmann et al. 2011). Correct stratification of OA vs. NOA patients is of utmost importance for the subsequent analysis of MC’s role in testicular immunopathology.
The term “Sertoli-cell-only phenotype (SCO)” (Wyrwoll et al., Nat Rev Urol 2024) should be used in Table 5 and throughout the text. The score used for semi-quantitative evaluation of spermatogenesis, and respective statistical analysis needs to be explained in the legend of Fig. 1 (in order to understand the meaning of Y axis scaling in A (“A” and “B” are missing in the graphs).
Atiaskin and colleagues did not address the query why sperm retrieval was only approx. 80% in OA and far below the expected approx. 50% in NOA (patient or sample selection bias?) (results, lines 212-).
As mentioned in the initial review, photomicrographs of representative histological overviews indicating the areas of interest used for further analysis need to be included in figs. 2-4 (so far only available in fig. 4 a).
In their rebuttal, Atiaskin and colleagues mention that they are planning (or already started) to perform multiplex immunohistochemical staining of MC tryptase, chymase and carboxypeptidase A3 (assumably, in the same set of NOA vs. OA testis specimens). To see these analyses in an extended and more comprehensive manuscript would definitely increase the impact of the work presented here.
Reviewer 2 Report
Comments and Suggestions for Authors
If authors agree, please make following amendments
1. Chang "m" to SE in line 172 on page 2
2. Chang "MS" to "MC" in line 255 on page 9.
2. Spell out "BL" in line 311 on page 11.
